# Ternary Blends for Self-Compacting Mortars Production Composed by Electric Arc Furnace Dust and Other Industrial by-Products

**DOI:** 10.3390/ma15155347

**Published:** 2022-08-03

**Authors:** Antonio López-Uceda, David Cantador-Fernández, Pedro Raposeiro Da Silva, Jorge de Brito, José María Fernández-Rodríguez, José Ramón Jiménez

**Affiliations:** 1Área de Ciencia de los Materiales e Ingeniería Metalúrgica, Ed. Leonardo Da Vinci, Campus de Rabanales, Universidad de Córdoba, 14071 Cordoba, Spain; p62louca@uco.es.es; 2Área de Química Inorgánica, Universidad de Córdoba, Avda. de la Universidad s/n, 14240 Belmez, Spain; p12cafed@uco.es; 3CERIS, Instituto Superior de Engenharia de Lisboa (ISEL-IPL), R. Conselheiro Emídio Navarro, 1, 1959-001 Lisbon, Portugal; silvapm@dec.isel.pt; 4CERIS, Instituto Superior Técnico, Universidade de Lisboa, Av. Rovisco Pais, 1049-001 Lisbon, Portugal; jb@civil.ist.utl.pt; 5Área de Ingeniería de la Construcción, Ed. Leonardo Da Vinci, Campus de Rabanales, Universidad de Córdoba, 14071 Cordoba, Spain

**Keywords:** self-compacting mortar, industrial by-products, electric arc furnace dust, ternary blends, mechanical strength, leaching tank test

## Abstract

This study is framed within the circular economy model through the valorisation of industrial by-products. This research shows the results of producing self-compacting mortars (SCMs) with electric arc furnace dust (EAFD) and other industrial by-products such as fly ash, conforming (FA) or not conforming (NcFA), from coal-fired power plants, or recovery filler (RF) from hot-mix asphalt plants. Three batches of SCMs, each with one industrial-by product (FA, NcFA, or RF), and three levels of EAFD ratio incorporation (0%, 10%, 20%), were tested. An extra batch with a greater amount of FA was manufactured. When the incorporation ratio of EAFD rose, the mechanical strength decreased, due to the presence of a calcium zinc hydroxide dihydrate phase; nevertheless, this decrease diminished over time. All SCM mixes, except the 40C 40FA 20 EAFD mix, were above 20 MPa at 28 days. All mixes named 70C and 40C reached 40 and 30 MPa, respectively, at 90 days. Mixes with EAFD showed less capillarity and no difference in water absorption by immersion with respect to mixes without EAFD after 91 days. The SCMs designed proved to be stable in terms of leaching of the heavy metals contained in EAFD, where all the hardened SCMs were classified as inert.

## 1. Introduction

The enormous amount of waste generated worldwide challenges global sustainability. Turning waste into a new raw material is one key way to achieve integrated ecological productivity in all economic sectors. This should avoid the negative impacts of waste disposal at the local level, such as landscape deterioration, and water and air pollution. These are two crucial and current demanding tasks for mankind. The construction sector could have a key role in incorporating waste in cement-based materials, such as concrete or mortar [1].

This industry is constantly evolving and searching for more efficient technologies that allow optimizing natural resources. Self-compacting concrete (SCC) has been one of the most important advances in construction materials, and its use has become widespread in recent years in civil works and buildings. These fluid mixes can be placed and compacted under their own weight, without vibration, segregation, blockage of coarse aggregate, bleeding, or paste exudation [2]. The coarse aggregate content in SCC is lower than that of conventional concrete, so part of the performance of SCC is defined by the behavior of the SCM phase [3]. The use of cement to improve fluidity increases the cost and CO_2_ emissions of SCC, so the use of mineral additions, such as fly ash, blast furnace slag, or filler, modifies the properties of the paste and improves the fluidity of the mix without increasing its cost and environmental impact [4]. Hence, the SCM phase is characterized by a high amount of powder content. Previous evaluations of the SCM to analyze the characteristics of SCC have been carried out by different authors [5,6]. SCM can also be used for rehabilitating and repairing reinforced concrete structures [7]. The incorporation to SCM of different raw materials such as dolomite powder [8], metakaolin and zeolite [9], pumice [10], or recycled waste materials, such as treated marine sediments [11], calcined foundry sand from the metal casting industry [12], fly ash [13,14], silica fume [15,16], and ground clay bricks [17], has been proven to be viable.

Globally, among the industrial by-products generated in the world with a powdery size that can be incorporated as additions in SCM production, fly ash (FA) is commonly used. After pulverizing coal is burned to produce electricity in coal-power plants, different coal combustion by-products are generated [18]. These by-products amount to approximately 780 million metric tonnes in the world per year [19]. FA is the one that accounts for the greatest quantity of the total, at 68% [20]. In Spain, the annual reports provided by Spanish electrical companies declare that around 60% of the FA is meant for the cement industry, as a supplementary cementitious material in concrete and in road construction [18]. FA must conform to the standards EN 450-1: 2013, EN 450-2: 2006, and EN 14227-4 to be used as cementitious materials in concrete. The part of this by-product that does not comply with the fineness criteria (UNE-EN 933-10: 2010), which establishes a maximum of 40% retained (in mass) in the 0.045 mm sieve, is also known as non-confirming fly ash (NcFA). In Spain, this is landfilled and ranges between 30 and 40% of the FA generated in the last decade [18]. This waste disposed at landfills can cause environmental issues [21,22]. Cheerarot & Jaturapitakkul [23] studied FA disposed of for 6, 12, and 24 months in landfills as pozzolanic material in cement replacement for mortars, in comparison with FA grinded with a finer size particle distribution. Fineness was found to be an important factor for compressive strength. Torres-Gómez et al. [18] studied the replacement of siliceous filler with NcFA in mortars, finding that there was an improvement in compressive strength. The studies on the incorporation of NcFA in SCC carried out by Esquinas et al. [24,25] have proved its feasibility in terms of durability and mechanical properties.

Another type of industrial by-product that can be incorporated in cement-based materials results from heating and drying the aggregates in hot-mix asphalt plants. This waste, hereinafter called recovery filler (RF), drops from the rotating drum along with combustion gases and is collected by means of baghouse filters to avoid being vented into the atmosphere. Lin et al. [26] studied the application of this waste; despite the fact that it comes from natural aggregates and as such, its properties are expected to be similar, they commented that some authors found diverging results in which the addition of RF in asphalt mixes led to the decline of some properties. Its reincorporation in the manufacturing hot-mix asphalt process is limited to 3–4% [27]. In 2017, in Europe, the production of hot and warm-mix asphalt was near 300 million tonnes, whereas in Spain it was 15.2. The generation of RF is estimated at 4% by weight of the total hot-mix asphalt production [27,28]. Martin et al. [28] studied the use of RF in SCC finding no signs of segregation or exudation, as well as satisfactory results regarding resistance to reinforcement corrosion by carbonation and compressive strength regarding the technical provisions and standards. Esquinas et al. [27,29] evaluated the behaviour of SCC with RF through mechanical properties and durability. Despite the fact that the mechanical properties of SCC with RF were slightly lower than the ones of a reference mix with silicious filler, drying shrinkage at early ages also presented lower values. Martinez-Echeverría et al. [30] also found a loss of mechanical properties in SCC with RF relative to the reference one. However, Shahidan et al. [31] found that compressive strength in SCC with RF presented greater values than SCC with natural aggregates.

On the other hand, the management of hazardous waste is one of the most complex current challenges, since incorrect management can have irreversible environmental impacts. World steel production in 2017 amounted to nearly 1.7 billion tonnes of crude steel, including carbon, stainless, and other alloys. The amount produced in electric furnaces (arc and induction) was above a quarter of the total. These figures represented the greatest numbers in the last decade [32]. One of the by-products generated in this industry is electric arc furnace dust (EAFD), which is generated as a result of the vaporization of molten iron with nonferrous metals, CO bursting bubbles, and the ejection and dragging of particles from the metal bath, slag, and other materials in the oven [33]. Its production ranges between 10 and 30 kg per tonne of steel for scrap smelting and direct reduced iron [34]. It is composed of heavy metals with leaching potential, which leads to negative impacts, such as Pb, Zn, Cd, Cr, and Ni [35]; because of this, it has been classified as hazardous waste in the European Waste Catalogue [36]. Therefore, its management implies a very high cost for steel mills since it cannot be deposited in landfills without prior treatment. The Spanish steelmaking industry generates an amount of 115,000 tonnes of EAFD per year [37]. Its particle size distribution matches the requirements for its use as filler. Different researchers have studied the influence of EAFD on mortars in terms of mechanical properties and leaching behaviour in the solid state [38,39,40,41]. In these studies, the heavy metals Zn, Se, Cd, Mo, and Pb, and the chloride anion were identified as the elements of EAFD that exceeded the limits to not be classified as hazardous waste. Additionally, it is said that the formation of the phase CaZn_2_(OH)_6_·2H_2_O in mortars with EAFD was found, and it inhibited hardening during the early stages of curing [40]. Based on this issue, Massarwe et al. [42] confirmed their hypothesis of the addition of EAFD as a set retarder in concrete, and Magalhães et al. [43] successfully studied the possibility of applying a pre-treatment with a NaOH solution to enhance strength gain. Santamaría et al. [44] studied the use of another type of by-product in steelmaking, called electrical arc furnace slag, which presents a particle size distribution similar to a regular fine aggregate, in SCM. In this study, the gain of compressive strength over time was postponed when this by-product was used. Other authors have studied the solidification/stabilization of EAFD with different matrix compositions. Salihoglu & Pinarli [45] studied the use of paste samples based on cement, lime, or both for EAFD S/S. Samples with 35% of cement and lime and 30% of EAFD presented optimum composition for minimizing leaching test results. Fernández-Pereira et al. [46] studied the EAFD S/S with geopolymers based on low calcium fly ash. Fares et al. [47] compared the usage of EAFD and different conventional binders, such as silica fume and FA, with Portland cement in mortars. One of the features of SCMs that make it adequate for the encapsulation of potential harmful components is the higher density of the cement matrix than in regular mortars [40]. This, along with the need for filler material in SCMs, makes EAFD and the other industrial by-products excellent candidates for this purpose.

This study analyses the influence of incorporating different industrial by-products with a powdery size, such as EAFD, RF, FA, and NCFA, as fillers on the performance of self-compacting mortars (SCMs). To the extent of the authors’ knowledge, none of the existing studies focused on the simultaneous use of FA, NcFA, or RF and EAFD as fillers in SCMs. This study addresses the topic regarding the use of ternary blends combining the use of EAFD and other industrial by-products, along with cement, to produce sustainable SCMs; this appears to be highly suitable due to its appropriateness as a filler and capability of limiting the release of potential pollutant elements. For this purpose, this investigation presents the characterization of the materials used, the performance of sustainable SCMs, and the leaching behaviour in the monolithic state. Therefore, this study contributes to the circular economy, since it avoids landfilling through its incorporation in SCMs and, consequently, reduces the consumption of raw materials.

## 2. Materials and Methods

### 2.1. Materials

For SCM production, two fractions of aggregates of siliceous nature were used: fine sand 0/1 (NS 0–1) and coarse sand 0/4 (NS 0–4). Figure 1 shows the particle sieve distribution of both sands. NS 0–1 and NS 0–4 presented water absorption after 24 h of 0.7% and 1.1% and specific gravity of 2.58 and 2.55, respectively. The cement used was CEM I-42.5 R (CEM) and it complied with NP EN 197-1. It had a specific gravity of 3.14. The high-performance/strong water reducer superplasticiser used, SikaPlast 898 (SP), is based on a combination of modified polycarboxylates in an aqueous solution that works by electrostatic and steric repulsion.

The different by-products used as fillers were conforming fly ash (FA), non-conforming fly ash (NcFA), recovery filler (RF), and electric arc furnace dust (EAFD). The FA was supplied by SECIL Company (Lisbon, Portugal) and complied with the NP EN 450-1 and NP EN 450-2 standards; on the contrary, the other FA used did not comply with the fineness standard UNE-EN 933-10: 2010, therefore named NcFA. It was obtained from the Puente Nuevo coal-fired power station (Córdoba, Spain) of Viesgo electric company. The origin of the RF was from waste powder from asphalt mixture sands supplied by PAMASA (Málaga, Spain). The hazardous waste, EAFD, came from a steelwork located in Zumárraga (Guipúzcoa, Spain) that uses an electric arc furnace (EAF) for steel production.

### 2.2. Test Methods for Materials Characterization

The chemical composition of CEM and EAFD was found by X-ray fluorescence (XRF) using 4 kW of power and S4PIONEER, BRUKER equipment, whereas for FA, NcFA, and RF it was done by analysis of the dispersive energy of X-rays (EDAX).

The mineralogical composition of the raw materials was determined by X-ray diffraction (XRD) using a Bruker D8 Discover A 25 with Cu Kα radiation (λ = 1.54050 A; tube voltage: 40 kV; Tube current 30 mA). For EAFD, goniometric scanning was used from 10° to 70° (2θ°) at a speed of 0.00625°/min, whereas for the rest of the materials, the speed was 0.05°/min. The identification of the main minerals was done by comparison with the JCPDS Powder Diffraction File database [48].

The specific surface area of the CEM, FA, NcFA, RF, and EAFD samples was analysed by the Brunauer-Emmett-Teller method (BET), using Micromeritics ASAP 2010 equipment, Norcross, GA, USA. The single-point pore volume was determined from the amount adsorbed at a relative pressure of ∼0.99. The real particle density was estimated according to UNE 80103:2013. The particle size distribution of CEM, FA, NcFA, RF, and EAFD was measured using a Mastersizer S 2000 device (Malvern Instruments, Malvern, UK) with a previous ultrasonic homogenization for sample particle dispersions. Furthermore, the adsorption-desorption isotherms of nitrogen were determined to study the pore size distribution in CEM, FA, NcFA, RF, and EAFD by means of the DFT (density function theory).

Thermogravimetric and differential thermal analysis (TGA-DTA) was performed in a Setaram Setsys Evolution 16/18 apparatus under nitrogen at a heating rate of 5 °C/min in FA, NcFA, RF, and EAFD samples.

The standard UNE-EN 12457-4:2003 [49] is a basic characterization procedure which was conducted on 0.090 kg of each filler (FA, RF, NcFA, and EAFD), by means of a single-step batch leaching test, in which the solution was shaken for 24 ± 0.5 h at an L/S ratio of 10 L/kg. After the contact phase, the samples were left to decant, then filtered and a subsample of 40 mL of eluate was collected for testing and analysed within 24 h, for several elements and/or anions using ICP-MS and ionic chromatography, respectively.

### 2.3. Mortar Mix Proportions 

The composition of the mortar mixes (Table 1) was designed following the Nepomuceno method [5], separating powders (CEM, FA, RF, NcFA, and EAFD) and fine aggregates (NS-0/1 and NS-0/4). The self-compacting parameters used based on the absolute volume proportions were: powder materials and fine aggregates (V_p_/V_s_), which were kept constant at 0.70 for all the mixes; and water and powder materials (V_w_/V_p_), with a value of 0.83 and 0.78 for mixes with 70% and 40% of cement with respect to powder materials, respectively. The mass of the superplasticizer and mass of the powder materials ratio (Sp/%p) were adjusted for each mix. This increased as the EAFD ratio incorporation rose since the EAFD had finer particle size distribution and greater specific surface area than the powders replaced (RF, FA, and NcFA).

The self-compacting parameter regarding the fresh state of SCM was determined through the *Gm*. It is obtained by means of the slump-flow test; a truncated cone with a height of 60 mm and diameters of 70 and 100 mm at the top and bottom, respectively. It was filled up and then it was immediately lifted to let the fresh mortar spread out on a plate with a smooth and dampened surface. The average of the two perpendicular diameters of the spread mortar was used in the formula for *Gm*, provided by Nepomuceno et al. [5], to obtain the *Gm* parameter (Equation (1)). This value decreased when the EAFD content increased, even with the addition of superplasticiser. Above the amount used in mixes with EAFD incorporation (Table 1), the phenomena of segregation and water exudation were found. Although the *Gm* values in some mixes were lower than what is recommended [5], none of the aforementioned phenomena occurred.
(1)$$Gm={\left(\frac{Dm}{D_{0}}\right)}^{2}-1$$
where *Dm* is the mean of the diameters of the spread mortar, in mm, and *D*_0_ stands for the initial diameter at the base of the cone, in mm.

The designation of the mortars depends on the composition by volume of the powders as shown in Table 1.

To produce the SCM specimens, a standard mortar mixer was used. The constituents in solid state were homogenised 30 s after 80% of the water was poured in the mixer. Two minutes later, the remaining 20% of the water was added with the superplasticiser previously diluted. Then, the mixer was stopped for 1 min to ensure that all the constituents were mixed and to clean the mixer-blades. Finally, the mix went on for another 1 min and prismatic specimens, 40 × 40 × 160 mm^3^, were cast without agitation or mechanical compaction. Before 24 h, 48 h, and 72 h for mix specimens with no EAFD, with 10% EAFD, and with 20% EAFD, respectively, were demoulded, and afterwards stored in a climatic chamber at 50 ± 5% relative humidity and 20 ± 2 °C. This increasing time between manufacturing and demoulding as EAFD percentage rose was due to the retardation of hardening of cement matrixes in the presence of EAFD, as aforementioned in the introduction section.

### 2.4. Test Methods for SCM

The mineralogical composition of the SCM was determined using the equipment used for raw materials with 0.05°/min of speed. For this purpose, samples were collected after testing compressive strength, and then immersed in ethylic alcohol and stored at least 72 h to inhibit any setting reaction and to dry by solvent replacement [6]. The XRD patterns of the crystalline mineral phases in SCM were compared with the JCPDS powder diffraction file database [50].

The mechanical properties of the SCM in hardened state were studied at 28 and 91 days of curing. The flexural strength was obtained using a standard three-point-bending test with a 100 mm span, and compressive strength was measured on perpendicular edges of each of the two residual pieces obtained from the flexural test, following the standard UNE-EN 1015-11:2000 [50]. The rate of loading for flexural and compressive strength were 0.04 kN/s and 0.4 kN/s, and their value corresponds to the average of the three tests.

The following physical properties related to the durability were studied: water absorption by immersion and capillarity, and dry bulk density; these tests were carried out after 91 days of curing in three specimens each. Water absorption by immersion was obtained by measuring the water absorbed after immersion until mass became constant, and its apparent volume was determined by subtracting the saturated mass and weighing the sample under water. To study the water absorption by capillary, the capillarity coefficient was obtained in accordance with UNE-EN 1015-18:2003 [51]. Dry bulk density was yielded in the relationship between the dry mass of the specimen and its apparent volume, following the standard UNE-EN 1015-10:2000 [52].

With the aim of analysing the capability of encapsulating heavy metals, with potential hazardous impacts to the environment, in SCM due to the presence of EAFD, the test conducted was the tank-leaching test according to XP X31-211:2012 [53]. It determines the leachability of a solid waste material, in this case generated in a solidification process. Prismatic sample halves were introduced in a PET container with a liquid to solid ratio of 10 with de-ionized water being agitated for 24 h. After that, the pH and electrical conductivity were measured and the amount of elements and anions was determined using ICP-MS and ionic chromatography, respectively.

## 3. Results and Discussion

### 3.1. Materials Characterization

Table 2 and Table 3 show the chemical composition of FA, NcFA, and RF by EDAX, and CEM and EAFD by XRF, respectively. Figure 2 shows the X-ray diffractograms of CEM, FA, NcFA, RF, and EAFD. The CEM sample presented a high content of larnite (Ca_2_SiO_4_, JCPDS no. 09-0351), hatrurite (Ca_3_SiO_5_, JCPDS no. 86-0402), gypsum (CaSO_4_·2H_2_O, JCPDS no. 33-0311) and to a lesser extent, components such as gismondinea ((Ca, Na_2_)Al_2_Si_2_O_8_·4H_2_O, JCPDS no. 21-0840) and calcite (CaCO_3_, JCPDS no. 80-0742) (Figure 2). The XRF analysis agreed with the mineralogical phase found in the diffractograms.

The XRD patterns of the NcFA and FA fillers (Figure 2) showed quartz (SiO_2_, JCPDS no. 33-1161) as the main phase; hematite (Fe_2_O_3_, JCPDS no. 33-0664) and mullite (Al_6_Si_2_O_13_, JCPDS no. 15-0776) were also found, the latter being in high quantity. The FA used presented almost identical chemical composition to the one used in da Silva et al.’s [6] research. The chemical composition determined by EDAX analysis of NcFA and FA (Table 2) implied that both samples exhibited a small amount of unburned coal particles and FA presented a greater quantity of Mg and Ca in carbonate forms. FA showed similar composition to the ones studied by Fernandez-Periera et al. [46]. According to the diffractogram, a greater presence of amorphous phase in NcFA than in FA was estimated, since a more dented hump was located between 15° and 35° [54,55]. Although the percentage of amorphous was not determined, the XRD pattern of RF filler showed a dolomitic nature (CaMg (CO_3_)_2_, JCPDS no. 36-0426) as the only phase presented. The XRD patterns of EAFD showed that it was mainly composed of franklinite (ZnFe_2_O_4_, JCPDS no. 22-1012) and zincite (ZnO, JCPDS no.36-1451). Silva Magalhães et al. [56] and Dutra et al. [57] also found these main phases in EAFD. To a lesser extent, other minor phases were found as sodium (NaCl, halite, JCPDS no. 05-0628) and potassium (KCl, sylvite, JCPDS no. 73-0380) chloride [58]. These results are in agreement with the chemical composition determined by EDAX (Table 3).

Table 4 shows the particle density (standard UNE- 80103: 2013), the specific surface area by Brunauer–Emmett–Teller (BET), and single-point pore volume (Vp) of CEM, FA, NcFA, RF, and EAFD. The high particle density value in EAFD is due to the higher density of the phases presented, franklinite and zincite, with a greater density of 5 g/cm^3^. The RF presented a surface much greater than the rest of materials, between 10 times the EAFD and 40 times the NcFA in terms of BET surface, and between 25 times the EAFD and 90 times the NcFA in terms of Vp.

Figure 3 shows the particle size distribution of CEM, FA, NcFA, RF, and EAFD in percentage. CEM, FA, and RF presented a similar distribution curve and maximum peak: for CEM 27 μm, for FA 39 μm, and for RF 32 μm. NcFA showed a trimodal distribution with a majority presence of coarser particles, with a maximum peak of 48 μm. The percentage of medium-sized particles, from 1 to 10 μm, is much lower than that of the rest of the samples, although the amount of particles smaller than 1 μm is similar, except for CEM where it is slightly greater than the rest. The EAFD sample presented a bimodal curve, medium-sized and small particle fraction, with very pronounced peaks at 2.6 µm and 0.3 µm respectively. Thus, it is a material with very fine particles.

Figure 4 shows the pore size distribution (nm) versus its volume and the cumulative curve, both in cm^3^/g. The size of the pores is classified by the International Union of Pure and Applied Chemistry (IUPAC) as follows: pores with sizes above 50 nm are named macropores, those ranging between 2 and 50 nm are mesopores, and those under 2 nm are micropores. Similar distributions were obtained in CEM, FA, and NcFA, with a higher concentration of pores in the range of small mesopores (2–10 nm). The maximum pore volume peak is around 3 nm in these samples. FA exhibited a small amount of micropores (between 1.7 and 2 nm), which did not happen with NcFA.

EAFD had a more uniform distribution throughout the range of pore sizes, although a large part of these pores were concentrated in the range of small mesopores. Micropores were also observed between 1.7 and 2 nm and its maximum peak in pore volume was around 3 nm in diameter.

The RF sample had no micropores and a few small mesopores. The largest number of pores was found between medium-large mesopores (20–50 nm) and macropores. The EAFD’s highest volume peak was near 31 nm, with a large dispersion of the curve. The RF presented the highest porosity, in agreement with the data in Table 2.

Figure 5 shows the thermogravimetric analysis (TGA) and differential thermal analysis (DTA) of FA, NcFA, RF, and EAFD. A loss of mass of around 3.2% occurs between 440 and 700 °C in FA, mainly due to the decomposition of calcium and magnesium carbonates and the formation of CO_2_ thanks to the combustion of unburned ones. This process shows in the thermal curve as an exothermic peak at around 600 °C. Fernández Pereira et al. [46] obtained a loss of mass of 3.32% at 750 °C in a FA sample. For NcFA, the loss was 1.5%, mainly due to the unburned carbonates. This is reflected in the peak of the DTA of NcFA, which is more defined than that of FA. The RF mass loss is around 48% (600–800 °C), and is due to the decomposition of the dolomite carbonate. The complexity of the EAFD makes it difficult to interpret its diagram. The changes in mass produced up to 650 °C are due to losses of hydration products and oxidative processes. After 650 °C, the loss of mass is due to sintering and melting of the chlorides present as sylvite and halite (Figure 2).

Table 5 shows the 12 heavy metal elements and the three anions determined by the waste acceptance criteria, EU LD 2003/33/EC [59], at landfills. This so-called European Landfill Directive classifies waste as hazardous, non-hazardous, and inert materials, according to the release levels. EAFD was classified as hazardous; the fluoride anion, Cu, Cr, Zn, Cd, and Hg heavy metals exceeded the limit for inert materials; Se, Mo, and the sulphate anion did so for non-hazardous limits; and Pb and the chloride anion for hazardous waste. The high release level of Pb in EAFD excludes its disposal at landfills without any treatment [60]. FA and NcFA were classified as non-hazardous materials due to the Cr, Mo, and sulphate anion content exceeding the corresponding limits. FA and NcFA release levels are in accordance with those exposed by Tsiridis et al. [61], who studied six different fly ash from Europe and in all of them Cr, Mo, and the sulphate anion exceeded the inert limit. RF was classified as inert, as expected. The fillers used (FA, RF, and NcFA) are not treated as potentially harmful to the environment, since its use as a new raw material construction is widely extended, unlike EAFD. 

### 3.2. Mineralogical Composition of the SCM

Figure 6, Figure 7, Figure 8 and Figure 9 show the mineralogical phases in SCM after 28 days of curing. The following phases were presented in all of the samples: calcite (CaCO_3_, JCPDS no.05-0586), ettringite (Ca_6_Al_2_(SO_4_)_3_(OH)_12_26H_2_O, JCPDS no. 72-0646), microcline (KAlSi_3_O_8_, JCPDS no. 19-0932), hatrurite (Ca_3_SiO_5_, JCPDS no. 86-0402), larnite (Ca_2_SiO_4_, JCPDS no. 09-0351), quartz (SiO_2_, JCPDS no. 33-1161), and portlandite (Ca(OH)_2_, JCPDS no. 04-0733). The presence of the last phase decreased when EAFD was incorporated because of the existence of heavy metals such as Cd, Ni, and Pb [62,63]; additionally, the phases calcium zinc hydroxide hydrate (CaZn_2_(OH)_6_·2H_2_O, JCPDS no. 25-1449), simonkolleite (Zn_5_(OH)_8_Cl_2_H_2_O, JCPDS no. 76-0922), and franklinite (ZnFe_2_O_4_, JCPDS no. 65-3111) were revealed. Franklinite (ZnFe_2_O_4_) comes from EAFD, as seen in Figure 2. The new phase detected, calcium zinc hydroxide hydrate (CaZn_2_(OH)_6_·2H_2_O), is a characteristic phase of the reaction of mortar with EAFD [38,40,61,64] identified at detriment of the portlandite phase. The phases quartz (SiO_2_), microcline (KAlSi_3_O_8_), and gismonde ((Ca, Na_2_) Al_2_Si_2_O_8_·4H_2_O, JCPDS no. 21-0840) correspond to the natural aggregate used.

### 3.3. Mechanical Strength

The compressive strength test results are presented in Figure 10. Firstly, regarding the mixes without EAFD, taking the 70C 30FA mix as the reference one, 70C 30NcFA showed a gain of 21.2% at 28 days of curing, whereas this gain was 10.7% at 91 days, and mixes with RF presented no loss or gain at the corresponding age. This result may be justified by the following reasons: the presence of coarser particles in NcFA (maximum distribution peak at 48 μm) than in FA and RF (maximum distribution peak at 39 μm and 32 μm respectively) (Figure 3), which led to a better skeleton filler with cement; the ratio mass of filler to mass of cement is lower in the 70C 30NcFA mix; or/and the admixture content has been lower in the 70C 30NcFA mix. The chemical composition and mineralogical phases presented in mortars with FA and NcFA fillers were alike, except the fact that NcFA presented a greater amount of amorphous phase, which could be another factor for justifying the higher compressive strength found. The 70C 30RF mix presented a loss of compressive strength of 4.2% relative to the reference mix (70C 30FA) at 28 days and 9.8% at 91 days, due to the non-pozzolanic nature of RF. The compressive strength values of 70C 30RF and 70C 30FA at 28 days were similar to the corresponding ones of Nguyen [8] and Türkel & Altuntas [65], respectively. 40C 60FA showed a decrease of 39.4% relative to the reference mix (70C 30FA) at 28 days, and 46.5% at 91 days due to the lower amount of cement used.

Secondly, the incorporation of EAFD in SCM resulted in a significant decrease in compressive strength, as expected. The compressive strength values at 28 days presented for all the mixes were above 20 MPa, except for the 40C 40FA 20EA mix (15.8 MPa); the minimum compressive strength established by the Spanish Structural Code for structural concrete with no reinforcement [66]. At 91 days, all mixes named as 70C surpassed 40 MPa, and the 40C 40FA 20EA mix reached more than 30 MPa. This structural code [66] allows the values of compressive strength at 91 days to be taken into account if the concrete does not bear any load until 3 months after placing. The loss of compressive strength at 28 days in mixes with 20% of EAFD incorporation relative to the corresponding mixes without EAFD, with the same filler, were 55.1%, 60.1%, 59.2%, and 53.0% for 70C 10FA 20 EA, 70C 10RF 20 EA, 70C 10NcFA 20 EA, and 40C 40FA 20EA, respectively. Vargas et al. [63] obtained a decrease of 20% of compressive strength at 28 days in mortars with 25% EAFD incorporation. In Ledesma et al.’s [38] research, the mechanical strength decrease as EAFD incorporation rose was justified by, among other reasons, the high amount of Zn that prevented the regular hydration of cement. This is in accordance with the finding of CaZn_2_(OH)_6_·2H_2_O instead of the portlandite phase in the mineralogical study of the SCM (Figure 6, Figure 7, Figure 8 and Figure 9). The gain of compressive strength over time in mixes with 20% of EAFD incorporation was greater than in mixes with no EAFD; these gains were 29.3%, 21.8%, 18.1%, and 14.0% for 70C 30FA, 70C 30RF, 70C 30NcFA, and 40C 60FA mixes; and 63.7%, 162.5%, 54.8%, and 90.0%, respectively. This has been reported by other authors [63,67] and has been attributed to the delay in the hydration of cement.

Despite the fact that the flexural strength (Figure 11) results presented similar trends to the compressive strength ones, when comparing mixes with no EAFD incorporation with the reference mix (70C 30FA) at 28 days, mixes with RF and NcFA presented a gain of 12.7% and 17.3%, respectively, and the 40C 60FA mix showed a drop of 25%. In the rest of the mixes, a lower reduction was found between the values with EAFD incorporation and those corresponding to the mixes without EAFD (with the same filler), at 44.5 ± 6.5%. At 91 days of curing, mixes with no EAFD presented similar values to those at 28 days, whereas mixes with EAFD presented a great gain of flexural strength. Conversely, Lozano-Lunar et al. [40] found that the flexural strength results in mixes without EAFD did not differ from mixes with a similar amount and nature of EAFD to this study. It has been reported [68] that an increase in compressive and flexural strength results from EAFD incorporation up to 15% in conventional concrete, relative to mixes without any addition. A linear regression between the flexural, *x* axis, and the compressive, *y* axis, strength values was carried out (intercept or constant value equals 0), and a strong correlation was found (R^2^ = 0.97). The compressive strength of all mixes resulted in 5.52 times the flexural strength; this is in agreement with da Silva et al. [43], who found an almost identical relationship for the same percentage of EAFD incorporation in mortars with cement and EAFD as binders.

### 3.4. Physical Properties Related to Durability

Table 6 shows the physical properties studied related to durability at 91 days of curing and the dry bulk density of the SCM produced. With regards to density, there were no relevant differences when comparing mixes with the same amount of cement and EAFD but different filler. As expected, the dry bulk density increased as the EAFD incorporation ratio rose, due to the greater particle density of EAFD, as seen in Table 4. Similarly, mixes with 40% of cement presented lower density relative to the corresponding ones with the same EAFD incorporation ratio and 70% of cement, since FA had a lower particle density than cement. The results of the water absorption by immersion at 91 days did not show significant differences among the mixes with 70% of cement, whereas mixes named 40C had slightly greater values. Regarding the capillarity coefficient, mixes with 70% of cement with EAFD and no EAFD were clearly presented into two groups, because of the strong influence of EAFD. The mixes with EAFD presented lower capillarity than the ones without EAFD. These results could be justified by the findings of Lozano-Lunar et al. [41], who found lower large capillaries in a mix with a similar amount of EAFD relative to the reference one. It has been also reported [69] that the incorporation of steel by-products with powder size can worsen, or at least not improve, the porous structure in the cementitious matrix, due to the capacity of these finer particles to fulfil voids.

### 3.5. Leaching Behaviour

Table 7 shows the results of carrying out the leaching tank test XP X31-211:2012 in SCM samples. The levels of Zn, Cd, and Hg were below the detection limit and fluoride contents were under 0.2 mg/L for all the samples; hence, not included in Table 5. Fernández-Olmo et al. [70] found that the release of Zn in EAFD solidified in mortars was at a minimum when the pH was above 8, which agrees with the results. According to the release of pollutant elements listed by the European Landfill Directive [59] and the legal limits imposed by this regulation, despite the EAFD incorporation, all the SCM samples were classified as inert regarding the Directive. In Lozano-Lunar et al.’s [40] study, mixes with a similar ratio binder/EAFD were also classified as inert, even though the EAFD incorporation in the SCM mortar increased the release levels of Cr, Cu, Se, Mo, Ba, and Pb elements, and the conductivity and pH as well. Despite the low value presented, Pb (which was the most critical heavy metal) release decreased in the 40C 50FA 10EA and 40C 40FA 20EA mortars relative to those with a higher amount of cement. Some researchers have found that Pb release in an alkaline medium, such as cement-based materials, could ease its mobility. Ledesma et al. [39] reported an increase in Pb release starting at 11.5 pH, and Salihoglu & Pinarli [45] above 11. This is in line with the low pH values obtained in this research.

## 4. Conclusions

The aim of this research was to produce self-compacting mortars (SCMs) with electric arc furnace dust (EAFD) and other industrial by-products, such as fly ash, conforming (FA) or not conforming (NcFA), from coal-fired power plants, or recovery filler (RF) from hot-mix asphalt plants, used as fillers. A wide physical-chemical characterization of these industrial by-products was conducted, as well as the determination of its influence on the mineral phases formed and mechanical properties of the hardened SCM. The leaching stability of the hardened SCM samples was also evaluated. The main conclusions were as follows:FA and NcFA presented similar composition in the elemental analysis, with a high presence of quartz. RF presented a dolomitic nature. EAFD presented mainly oxides of zinc and/or iron. The maximum percentage of particle size of CEM, FA, and RF ranged between 27 and 39 μm, whereas RF presented the greatest value (48 μm). EAFD distribution particles presented a maximum size near 20 μm and two peaks at 2.6 µm and 0.3 µm. Regarding the pore size distribution, RF presented large mesopores and the others small mesopores and/or micropores.To produce SCM with similar self-compacting characteristics in the fresh state, EAFD incorporation required much higher superplasticiser contents to keep constant the water to powder volume ratio. This is due to the greater content in particles with a size below 1 µm.In terms of mechanical properties, the EAFD incorporated implied a decrease as the incorporation increased. The appearance of the phase CaZn_2_(OH)_6_·2H_2_O in SCM with EAFD at the expense of the portlandite phase caused a reduction of the mechanical strength of SCM. This detrimental effect was reduced over time. Nevertheless, all SCM mixes produced, except for the 40C 40FA 20 EAFD mix, were above 20 MPa at 28 days, which is the minimum established by the Spanish Structural Code for structural concrete with no reinforcement. This structural code allows the use of the compressive strength at 90 days if no load is applied on concrete after placing; all mixes named 70C reached 40 MPa, and the ones named 40C reached 30 MPa at that age, hence, the SCM produced could be suitable for higher requirements than at 28 days if the previous condition is set.The study on the physical properties at 91 days of curing showed that the incorporation of fillers FA, NcFA, and RF did not significantly affect their performance. Regarding EAFD incorporation, these properties were not negatively affected. In fact, the capillarity was lower for mixes with EAFD incorporation, which could be attributed to its finer particle size.The leaching test of SCM in the monolithic state did not show a relevant release of any potentially harmful element. All SCMs with EAFD produced were classified as inert, according to criteria established by the European Landfill Directive. The ternary blend mixes with lower amounts of cement (40%) and greater amounts of FA performed even better than with greater amounts of cement. These results imply the feasibility of increasing the amount of filler in SCMs with EAFD, in terms of EAFD harmfulness encapsulation for future research.

The findings of this study show viable solutions of combining different industrial by-products as new secondary raw fillers to produce ternary blends for cement-based materials such as SCMs.

## Figures and Tables

**Figure 1 materials-15-05347-f001:**
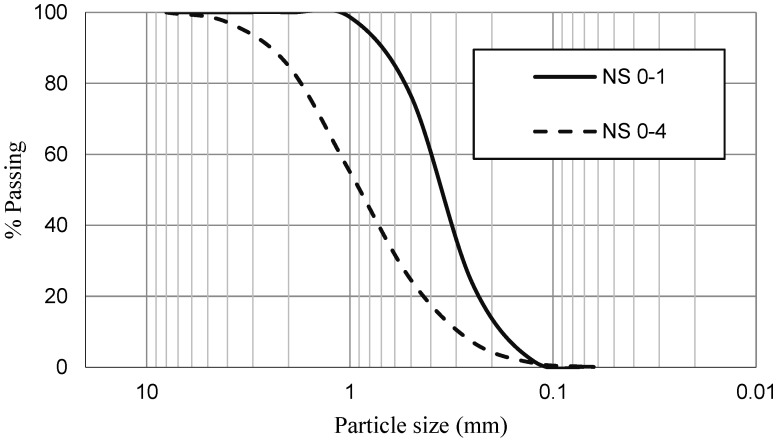
Particle size distribution of NS 0–1 and NS 0–4.

**Figure 2 materials-15-05347-f002:**
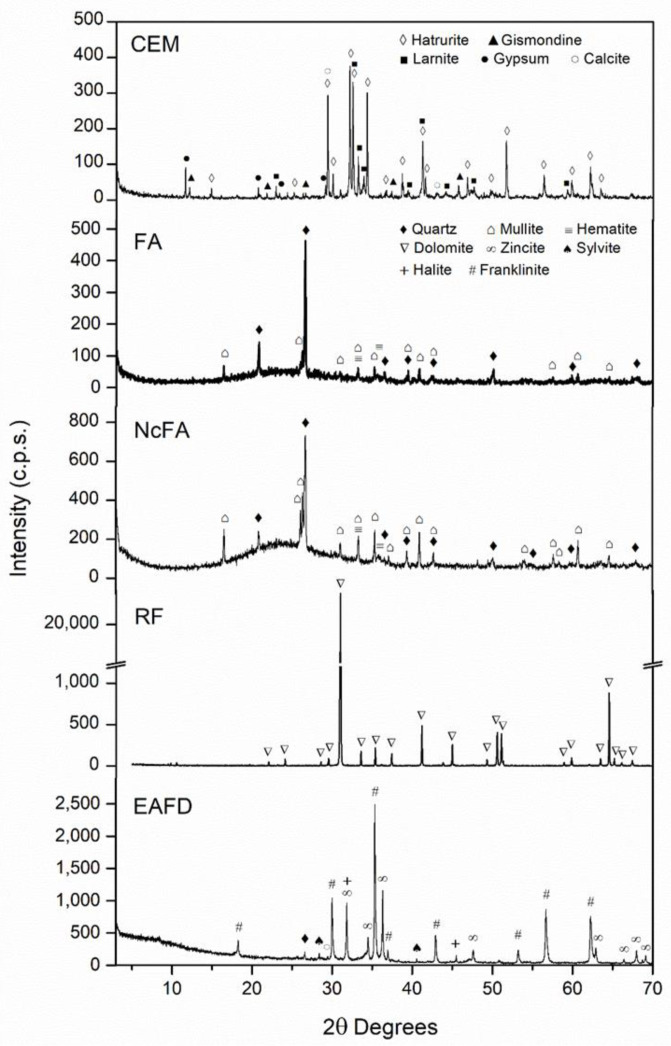
XRD patterns of CEM, FA, NcFA, RF, and EAFD.

**Figure 3 materials-15-05347-f003:**
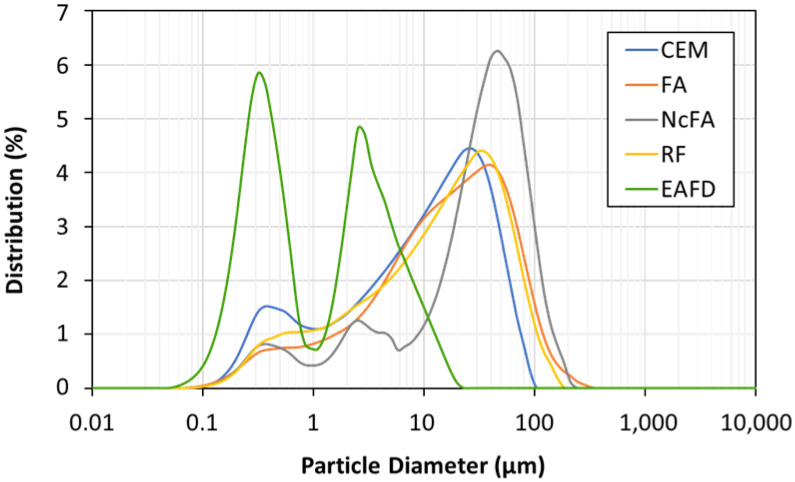
Particle size distribution of CEM, FA, NcFA, RF, and EAFD.

**Figure 4 materials-15-05347-f004:**
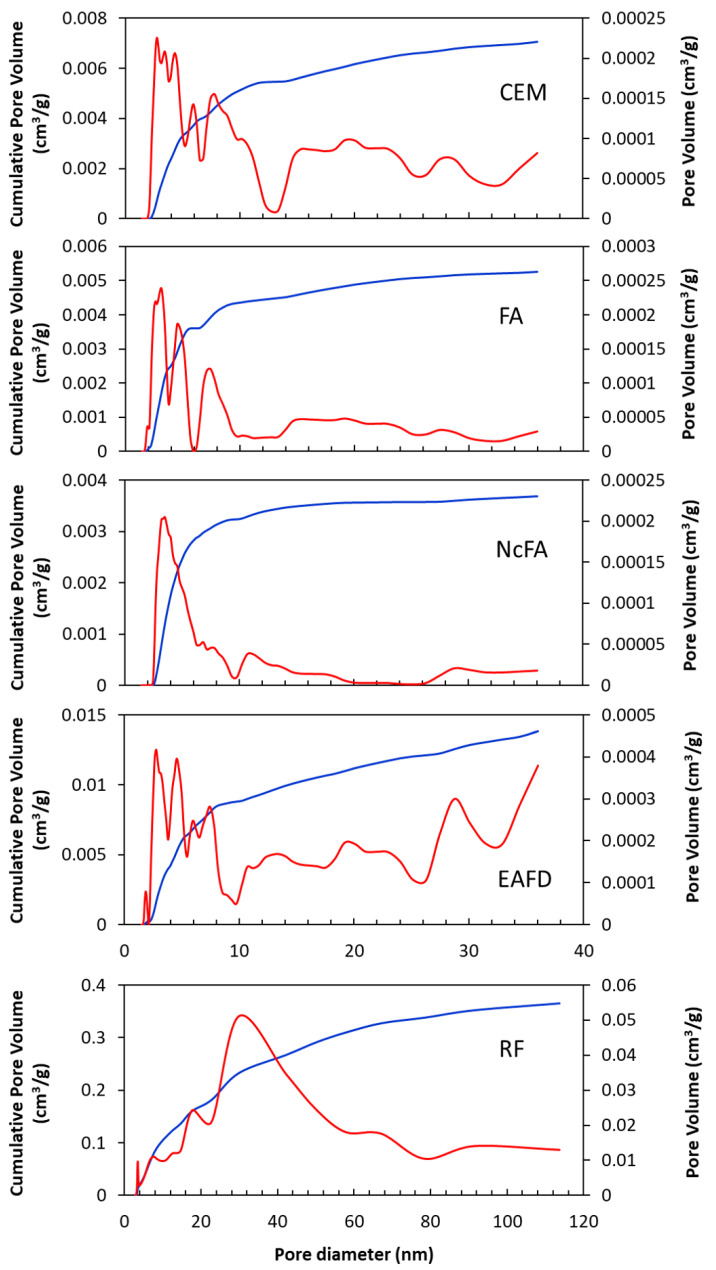
Pore size distribution (red line) and cumulative pore volume (blue line) of CEM, FA, NcFA, EAFD, and RF. *Note: X-axes for CEM, FA and NcFA are equal to EAFD’s one*.

**Figure 5 materials-15-05347-f005:**
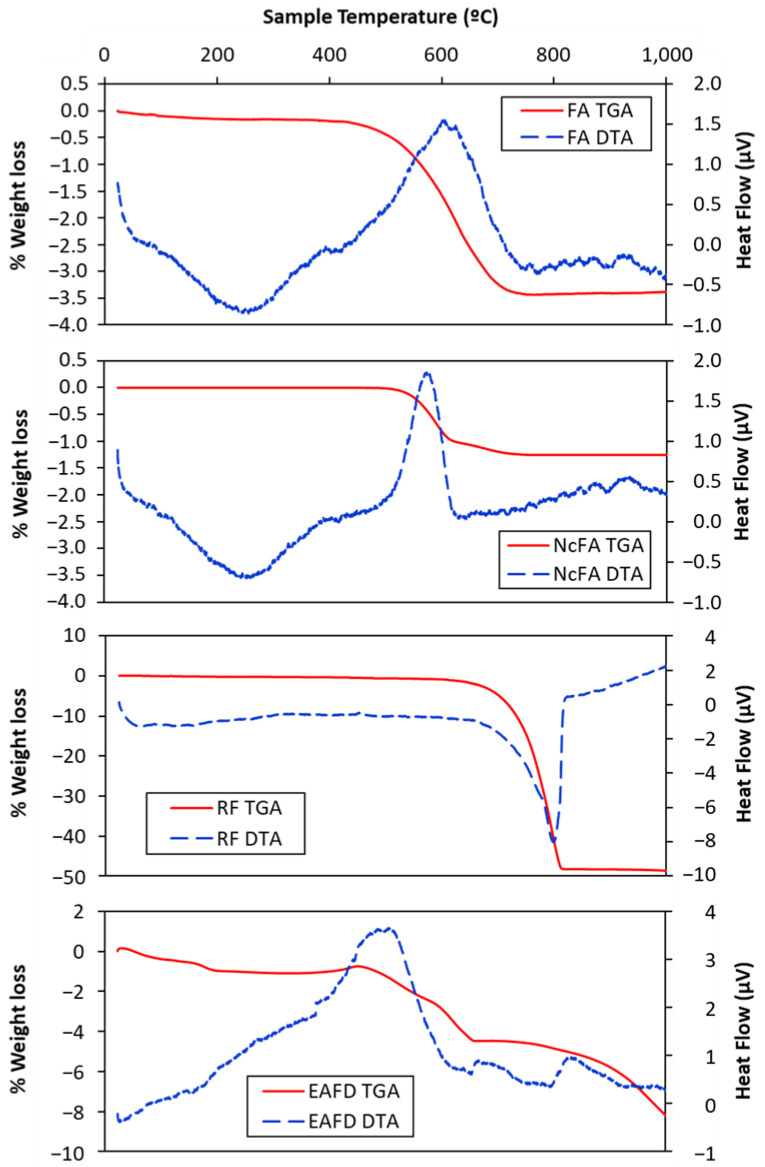
Thermogravimetric and differential thermal analysis of FA, NcFA, RF, and EAFD.

**Figure 6 materials-15-05347-f006:**
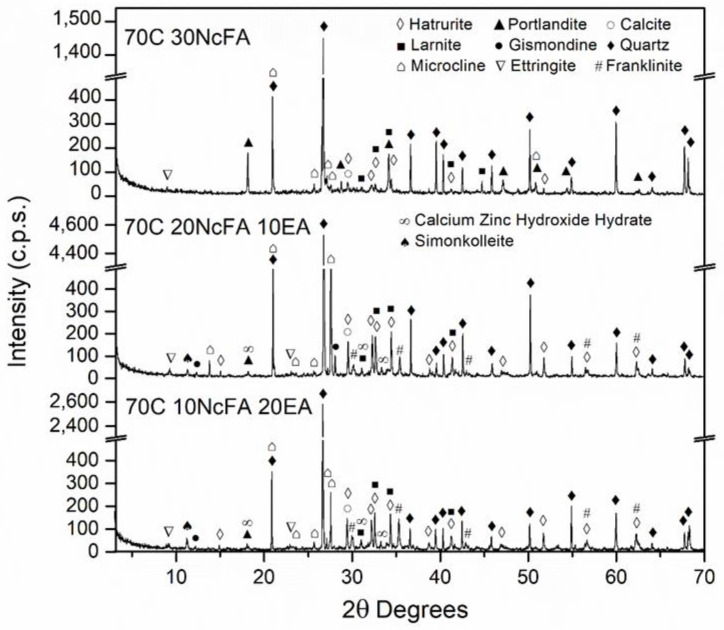
Mineralogical phases in 70C 30NcFA, 70C 20NcFA 10 EA and 70C 10N 20EA SCM mixes at 28 days of curing.

**Figure 7 materials-15-05347-f007:**
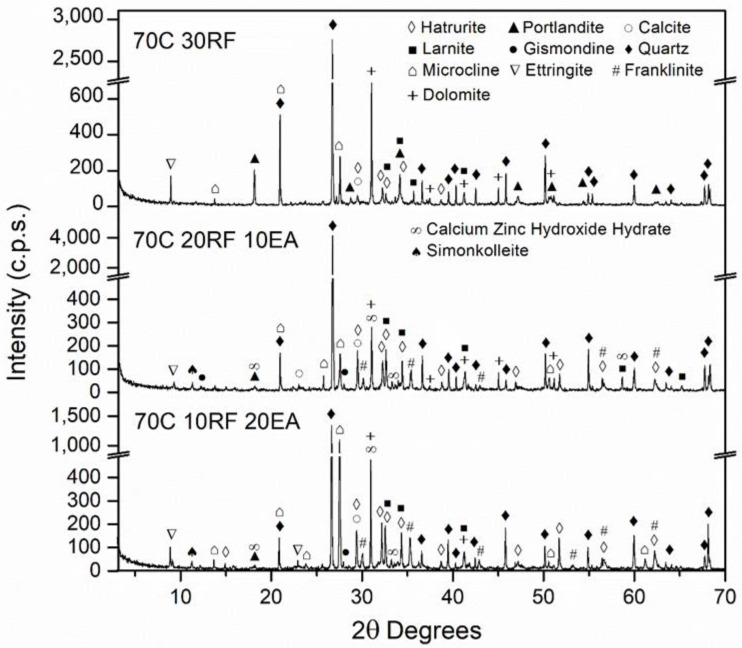
Mineralogical phases in 70C 30RF, 70C 20RF 10 EA and 70C 10RF 20EA SCM mixes at 28 days of curing.

**Figure 8 materials-15-05347-f008:**
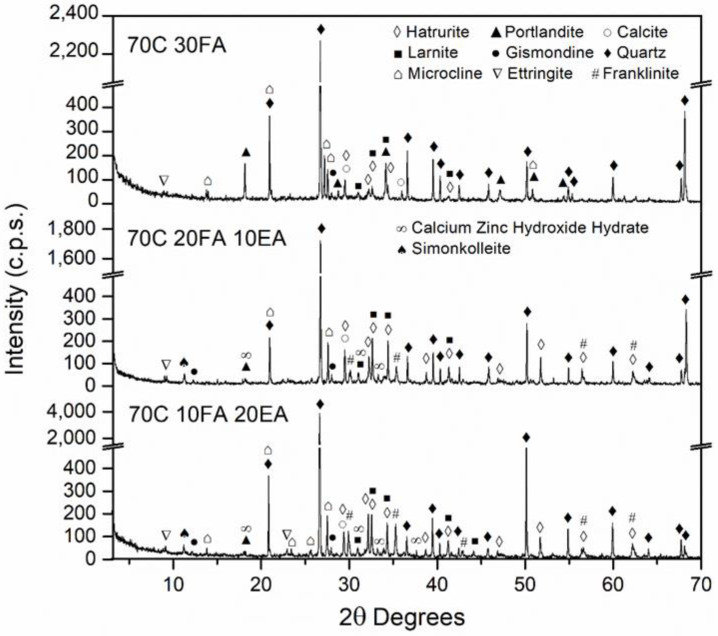
Mineralogical phases in 70C 30FA, 70C 20FA 10 EA and 70C 10FA 20EA SCM mixes at 28 days of curing.

**Figure 9 materials-15-05347-f009:**
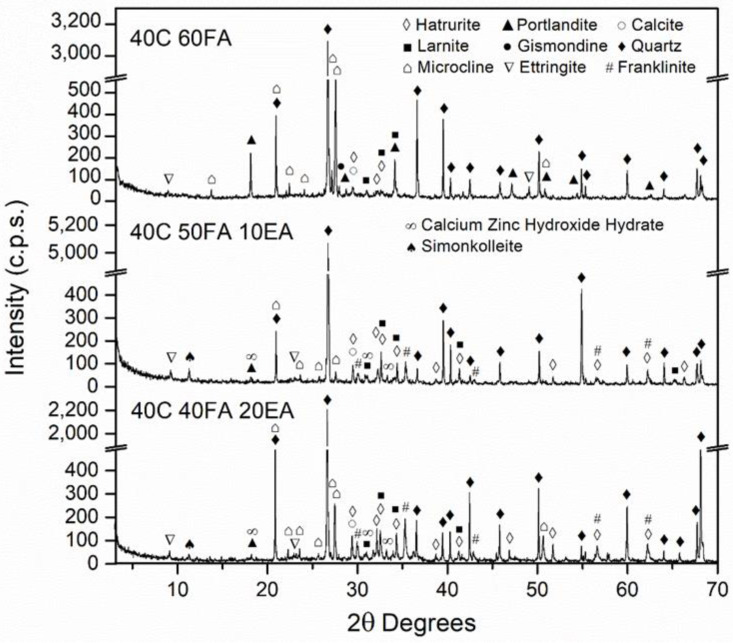
Mineralogical phases in 40C 60FA, 40C 50FA 10 EA and 40C 40FA 20EA SCM mixes at 28 days of curing.

**Figure 10 materials-15-05347-f010:**
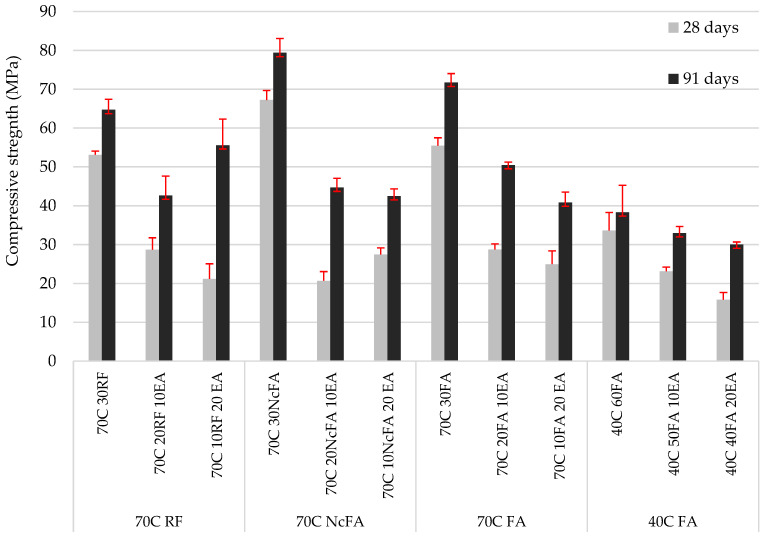
Compressive strength at 28 and 91 days.

**Figure 11 materials-15-05347-f011:**
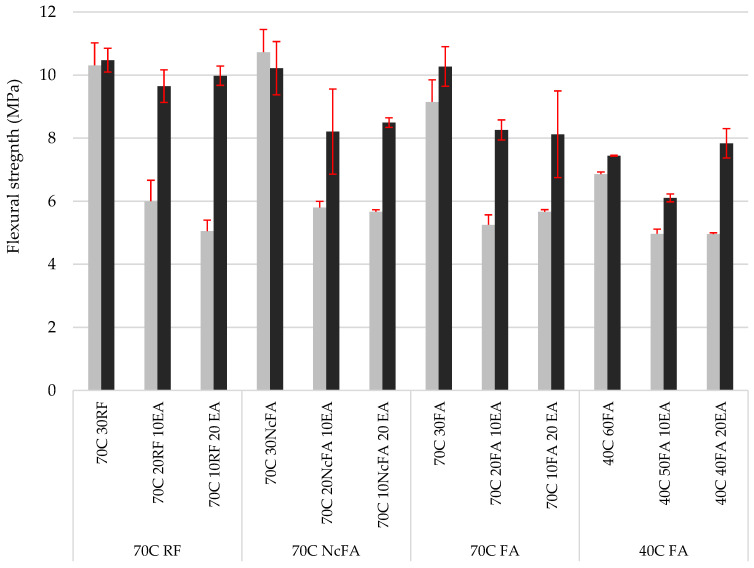
Flexural strength at 28 and 91 days.

**Table 1 materials-15-05347-t001:** Composition of the mortar mixes in kg/m^3^ and self-compacting parameters (% by volume).

Mixes	NS-0/2		NS-0/4		C		FA		NcFA		RF		EAFD		Sp	Water	V_w_/V_p_	Sp/%p	*Gm* ^1^
	kg/m^3^	%	kg/m^3^	%	kg/m^3^	%	kg/m^3^	%	kg/m^3^	%	kg/m^3^	%	kg/m^3^	%	kg/m^3^	kg/m^3^			
70C 30FA	558.2	21.7	549.5	21.6	670.5	21.4	215.0	9.2	-	-	-	-	-	-	6.5	253.2	0.83	0.73	5.45
70C 30NcFA	562.3	21.8	555.8	21.8	670.7	21.4	-	-	170.3	9.2	-	-	-	-	6.1	253.3	0.83	0.73	5.35
70C 30RF	561.9	21.8	555.4	21.8	670.2	21.3	-	-	-	-	263.5	9.1	-	-	6.8	253.1	0.83	0.73	5.03
70C 20FA 10 EA	562.1	21.6	555.6	21.6	670.2	21.2	143.3	6.1	-	-	-	-	117.4	3.0	13.6	253.1	0.83	1.47	5.53
70C 20NcFA 10EA	562.1	20.7	555.5	20.7	670.4	20.3	-	-	113.5	10.1	-	-	117.4	2.9	13.4	253.1	0.83	1.50	5.47
70C 20RF 10EA	567.2	21.8	560.6	21.8	676.5	21.4	-	-	-	-	177.3	6.1	118.5	3.1	16.2	246.2	0.83	1.43	4.31
70C 10FA 20 EA	561.8	21.6	555.2	21.6	670.0	21.2	71.6	3.0	-	-	-	-	234.7	6.0	16.5	253.0	0.83	1.71	4.43
70C 10NcFA 20EA	561.8	21.8	555.3	21.8	670.1	21.4	-	-	56.7	2.2	-	-	234.7	6.1	13.7	253.0	0.83	1.70	4.41
70C 10RF 20EA	567.1	21.8	560.5	21.8	676.3	21.3	-	-	-	-	88.6	3.0	236.9	6.1	16.6	246.2	0.83	1.69	4.02
40C 60FA	571.8	22.2	565.1	22.2	389.7	12.4	437.5	18.6	-	-	-	-	-	-	4.8	242.0	0.78	0.58	6.70
40C 50FA 10EA	571.6	22.0	565.0	22.0	389.6	12.3	364.5	15.4	-	-	-	-	119.4	3.1	11.5	241.9	0.78	1.32	6.18
40C 40FA 20EA	571.5	22.0	564.8	22.0	389.5	12.3	291.5	12.3	-	-	-	-	238.8	6.2	14.3	241.9	0.78	1.57	4.41

^1^ Relative spread area.

**Table 2 materials-15-05347-t002:** Chemical composition of FA, NcFA, and RF (EDAX).

Compound (% Weight)	FA	NcFA	RF
MgO	1.78	1.28	44.15
Al_2_O_3_	26.83	32.47	-
SiO_2_	55.44	54.54	-
CaO	2.93	1.38	55.85
Fe_2_O_3_	7.33	8.24	-
Balance CO_2_	5.69	2.10	0.00
Total	100.00	100.00	100.00

**Table 3 materials-15-05347-t003:** Chemical composition of CEM and EAFD (XRF).

Compound (% Weight)	CEM	EAFD
Na_2_O	0.11	2.59
MgO	1.76	1.67
Al_2_O_3_	4.59	0.94
SiO_2_	15.23	2.65
P_2_O_5_	0.03	0.2
SO_3_	4.05	2.21
Cl	0.05	11.37
K_2_O	0.62	1.96
CaO	54.71	2.93
TiO_2_	0.28	0.07
V_2_O_5_	0.02	-
MnO	0.04	2.51
Fe_2_O_3_	2.44	30.48
CuO	0.02	-
ZnO	0.03	32.77
SrO	0.04	-
Cr_2_O_3_	-	0.5
BaO	-	0.03
PbO	-	2.23
F	-	0.43
SnO_2_	-	0.08
Br	-	0.07
CdO	-	0.04
NiO	-	0.02
MoO_3_	-	0.01
Balance CO_2_	15.98	4.24
Total	100.00	100.00

**Table 4 materials-15-05347-t004:** Particle density, BET surface, and Vp of CEM, FA, NcFA, RF, and EAFD.

Properties	CEM	FA	NcFA	RF	EAFD
Particle density (kg/m^3^)	3140	2350	1860	2880	3810
S_BET_ (m^2^/g)	4.3	4.3	1.9	76.0	7.6
V_p_ ^1^ (cm^3^/g)	0.007	0.005	0.004	0.357	0.014

^1^ DFT method.

**Table 5 materials-15-05347-t005:** Leached concentrations of fillers (mg/kg) according to UNE-EN-12457-4:2003 standard and waste acceptance criteria EU LD 2003/33/EC [59].

					Criteria EU LD 2003/33/EC (L/S = 10)		
	FA	NcFA	RF	EAFD	Inert	N-Hazar. ^1^	Hazar. ^2^
Cr	3.59	2.68	n.d.	1.97	0.5	10	70
Ni	0.01	n.d.	n.d.	0.05	0.4	10	40
Cu	n.d. ^3^	0.02	n.d.	2.16	2	50	100
Zn	0.01	n.d.	0.01	24.05	4	50	200
As	n.d.	n.d.	n.d.	0.05	0.5	2	25
Se	n.d.	n.d.	n.d.	2.76	0.1	0.5	7
Mo	1.14	0.95	0.02	20.49	0.5	10	30
Cd	n.d.	n.d.	n.d.	0.14	0.04	1	5
Sb	n.d.	n.d.	0.02	n.d.	0.06	0.7	5
Ba	7.94	5.86	0.27	6.94	20	100	300
Hg	n.d.	n.d.	n.d.	0.18	0.01	0.2	2
Pb	0.02	0.01	n.d.	5480	0.5	10	50
Fluoride	3.7	<2.0	<2.0	65.8	10	150	500
Chloride	64	37	<2.0	24,100	800	15,000	25,000
Sulphate	4090	2879	24	16,300	1000	20,000	50,000
e.c. (µS/cm)	5720	3560	40.4	8560	-	-	-
pH	9.64	8.35	9.41	13.28	-	-	-

^1^ Non Hazardous; ^2^ Hazardous; ^3^ non detected.

**Table 6 materials-15-05347-t006:** Physical properties related to durability measured at 91 days of curing.

Mixes	Dry Bulk Density		Capillarity Coefficient		Water Absorption by Immersion	
	(g/cm^3^)		kg/(m^2^·min^0.5^)		(%)	
	Mean	*SD*	Mean	*SD*	Mean	*SD*
70C 30FA	1.769	0.001	0.211	0.001	9.87	0.13
70C 30NcFA	1.728	0.000	0.205	0.014	10.69	0.27
70C 30RF	1.764	0.004	0.188	0.038	10.00	0.17
70C 20FA 10 EA	1.843	0.005	0.126	0.008	9.99	0.13
70C 20NcFA 10EA	1.827	0.002	0.116	0.008	10.00	0.09
70C 20RF 10EA	1.858	0.005	0.111	0.003	9.58	0.13
70C 10FA 20 EA	1.903	0.020	0.133	0.009	10.50	0.13
70C 10NcFA 20EA	1.888	0.008	0.116	0.008	9.86	0.09
70C 10RF 20EA	1.920	0.008	0.120	0.001	9.87	0.01
40C 60FA	1.672	0.015	0.233	0.017	10.06	0.14
40C50FA 10EA	1.820	0.025	0.202	0.008	11.64	0.43
40C 40FA 20EA	1.849	0.001	0.193	0.006	11.03	0.05

**Table 7 materials-15-05347-t007:** Leaching task test results in hardened SCM.

Mixes	Cr	Cu	As	Se	Mo	Sb	Ba	Pb	SO_4_^=^	Cl^−^	pH	e.c.
	(mg/kg)								(mg/L)			(µS/cm)
70C 30FA	0.073	n.d.	0.002	0.005	0.024	0.014	0.128	0.005	8.0	<2.0	10.8	726
70C 20FA 10EA	0.106	0.007	0.001	0.014	0.112	0.003	0.115	0.030	17.4	11.0	11.1	820
70C 10FA 20EA	0.143	0.001	n.d.	0.022	0.256	0.005	0.154	0.124	23.3	32.0	11.4	888
70C 30RF	0.088	n.d.	n.d.	0.002	0.024	0.001	0.071	n.d.	11.5	<2.0	10.9	604
70C 20RF 10EA	0.101	0.004	n.d.	0.016	0.113	0.002	0.060	0.017	13.6	11.0	11.2	543
70C 10RF 20EA	0.172	n.d.	0.002	0.017	0.251	0.018	0.118	0.121	26.8	37.0	11.2	826
70C 30NcFA	0.080	n.d.	0.001	n.d.	0.023	0.004	0.046	0.001	9.9	<2.0	10.8	504
70C 20NcFA 10EA	0.098	0.003	0.001	0.023	0.067	0.002	0.036	0.043	10.1	7.0	11.0	425
70C 10NcFA 20EA	0.150	0.001	0.001	0.022	0.271	0.005	0.061	0.191	35.1	31.0	11.3	712
40C 60FA	0.088	n.d.	n.d.	n.d.	0.054	0.021	0.039	0.001	12.1	<2.0	10.3	280
40C 50FA 10EA	0.138	0.004	0.001	0.027	0.229	0.004	0.034	0.002	13.9	15.0	11.0	516
40C 40FA 20EA	0.158	0.001	n.d.	0.036	0.466	0.005	0.054	0.008	29.9	58.0	11.1	544

e.c. electrical conductivity.

## Data Availability

The data used to support the findings of this study are available from the corresponding author upon request.

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
