# Peer review of "Ternary Blends for Self-Compacting Mortars Production Composed by Electric Arc Furnace Dust and Other Industrial by-Products"

_materials, 2022, doi:10.3390/ma15155347_

Round 1
Reviewer 1 Report
The proposed work focuses on Ternary blends for self-compacting mortars production composed by electric arc furnace dust and other industrial by products. It is of potential interest to Materials journal readers.
Despite the importance of the subject addressed, this work needs many improvements to be ready for the publication in the Materials journal.
Specific points of improvement :
- The objective of this research must be more developed.
- Quality of figures must be improved.
- All test standards must be indicated in the manuscript.
Reviewer 2 Report
This is a very interesting study. The author observed the change of the compressive strength of mortar at 91d, and the characterization work is well done, but I suggest adding microscopic observation and analysis when studying the compressive strength.
Reviewer 3 Report
The research presents a comparative study between self-competed mortar mixes with different percentages of pozzolanic materials (FA, NcFA & RF) and new filling material (EAFD). The study considered three points of views, strengths (compressive & flexural), physical properties (dry bulk density, capillarity coefficient & water absorption) and environmental properties (Leaching behavior). The research concluded that the new filling material (EAFD) has negative impact on mortar strengths and has no impact on both physical & environmental properties.
The research is well organized and written in good English, however the following comments must be considered before acceptance:
1. Reference [48] is not mentioned in the manuscript
2. In page 22 line 474, Reference [64] is repeated twice
3. The abstract must contains the gap and some numerical outcomes
4. Numbers on horizontal axis if Fig. 4 is missing for CEM, FA & NcFA
5. Fig. 11 is a copy of Fig. 10
6. Present the Physical properties in Table 5 graphically
7. Present the Leaching task test results in Table 6 graphically
8. Conclusions must present numerical values to compare the performance of the tested mixes.
Reviewer 4 Report
The manuscript entitled ˝Ternary blends for self-compacting mortars production composed by electric arc furnace dust and other industrial by-products˝ is interesting and important in terms of waste recycling, however, significant improvement is needed before publication.
Here are some comments and suggestions to improve the manuscript:
INTRODUCTION:
Line 75 and 81: Please unify the labelling, e.g. NcFA/NCFA
Line 81: What was the reason for the improvement of compressive strength?
Line 117: Please add the space Ni[35].
Line 145: I think this sentence is a bit too strong: ˝This could contribute to a new paradigm of the circular economy...´˝
MATERIAL AND METHODS:
Lines 158, 159, 173: NS 0-1 or NS 0-2? Which is the correct label? Please check in further text as well.
Line 168: Please add the producer of NcFA.
Lines 174-181 are the same as 157-163 and lines 181-188 are the same as 164-171. Please remove this.
Line 230: Explain the abreviation Gm and how did you calculate this?
Table 1: How the mix designs were selected? Why did you choose 70 or 40 wt% of C? Why 30 or 60 wt% of other additives were added? Did you perform any preliminary tests? Please explain.
Lines 280-287: Did you perform a granular leaching test (compliance test) to see what happens after its lifecycle? I think the leaching concentrations will be much higher...
Line 286: What do you mean by the term ˝were extracted˝?
RESULTS:
Table 2 and 3: Please add the information about LOI.
Line 299: Nc-FA? Please unify.
Line 302: Please add the space in[6].
Line 304: particles instead of paticles
Line 308: What this mean? Or is this a mistake and you mean XRD?
Line 313: y, is this correct?
Figure 2: There is no data about the amorphous phase, please add this. These are diffractograms, not XRD. XRD is a technic to obtain data.
Line 319: Please check the methodology part. You used a different approach for adsorption-desorption.
Line 322: Do you know the approximate proportion of franklinite and zincite in the EAFD? Can you add this?
Line 353-354: ...so the BJH method was performed. This part is not clear. Please explain.
Figures 6-9: There is no information about the amorphous phase of SCM. Which phase increased/decreased by adding different raw materials? Please explain more in detail the addition of each precursor.
Line 409-431: Please check the font size.
Figure 11: Something is not ok with this graph. Please check. This graph is the same as graph 10.
Table 6: This is relatively high conductivity. How do you measure leachates? Did you acidify the solution and measure by ICP-MS after that?
Did you perform the compliance test for leaching of granular waste? in the end, all materials will probably end at the landfill, not as monoliths...Could you add the results of leaching for EAFD and other raw materials?
Round 2
Reviewer 4 Report
The authors improved the manuscript, however, few things are still missing:
-Table 2 and 3: Could you provide the literature for ˝Balance CO2˝? How did you calculate this? And please provide the data for LOI (loss on ignition).
-Table 5: Please add the values for inert, non-hazardous and hazardous waste in the table so that the reader can immediately see and compare the precursor data with the legislation values. And please provide the values in Table consistently, e.g. you can round the number 5483.87 to 5480.
